



# Residual Mean Circulation and Temperature Changes during the Evolution of Stratospheric Sudden Warming Revealed in MERRA

Byeong-Gwon Song and Hye-Yeong Chun

Department of Atmospheric Sciences, Yonsei University, Seoul, 03722, South Korea

*Correspondence to*: Hye-Yeong Chun (chunhy@yonsei.ac.kr)

**Abstract.** Residual mean circulation and temperature changes during the evolution of major stratospheric sudden warming (SSW) are investigated by composite analyses of 22 SSW events from 1979 to 2012 during the Northern Hemisphere winter (November–March) using four reanalysis data sets (MERRA, ERA-Interim, NCEP-NCAR, and JRA-55). The SSW events are classified as Type-1 or Type-2 based on the relative amplitude of planetary waves with zonal wavenumbers 1 and 2. The
residual mean circulation induced by each forcing term in the Transformed Eulerian mean (TEM) momentum equation and the temperature advection associated with the circulation are calculated for both types of SSW, based on the generalized downward control principle using the MERRA data set. When 'Lag = 0' is defined as the day on which the wind reversal occurred at 60° N and 10 hPa, strong poleward and downward motion exists at Lag = –8 and Lag = –1 for Type-1 and at Lag = –3 for Type-2, which is induced primarily by the Eliassen–Palm flux divergence forcing (EPD). The poleward and
downward motion is stronger for Type-2 than for Type-1. Gravity wave drag (GWD) produces a smaller contribution to the residual circulation than EPD. During the warming phase (at Lag = –2), strong temperature advection by the EPD induces primarily polar stratospheric warming. On the other hand, during the temperature recovery phase (at Lag = +2), anomalous negative temperature advection and diabatic cooling produce negative temperature tendency anomalies. Structures in the temperature tendency and temperature advection calculated using the MERRA data set are similar to those calculated using
the ERA-Interim data set.

## 1 Introduction

Stratospheric sudden warmings (SSWs) are one of the most dramatic events in the high-latitude stratosphere and are associated with the sudden breakdown of the polar vortex and a rapid increase in temperature. They were first observed using radiosonde by Scherhag (1952). SSWs usually occur in the northern hemisphere, greatly affecting circulation in the
northern winter stratosphere (Holton, 1980). Because temperature changes and associated wind reversal in the stratosphere during SSWs can spread downward, SSW events are important not only in the stratospheric circulation but also in the tropospheric weather (Baldwin and Dunkerton, 2001). Thus, various studies, including theoretical, observational, and numerical modeling of SSWs, have been extensively performed by many scientists.

Various definitions of SSW have been suggested recently (Butler et al., 2015). According to Charlton and Polvani
(2007) (hereafter CP07), one of the most commonly used definitions, SSW occurs when zonal-mean zonal wind at 60° N and 10 hPa is reversed from westerly to easterly. This definition is based on the theoretical study of Matsuno (1971) that suggested that upward propagating planetary-scale quasi-stationary waves are broken when they meet the zero wind in the stratosphere, which induces stratospheric warming due to downward motion produced by this wave forcing. In addition to the definition using zonal-mean zonal wind, other definitions using the meridional gradient of zonal-mean temperature, a
polar vortex geometry, and a Northern Annual Mode (NAM) index have been proposed as well (Labitzke, 1981; Waugh and Randel, 1999; Martineau and Son, 2013). Recently, Butler et al. (2015) documented the sensitivity of the frequency of SSW occurrence to the definitions of SSW using the NCEP-NCAR (Kalnay et al., 1996), ERA-40 (Uppala et al., 2005), and ERA-Interim (Dee et al., 2011) data set.



SSW events have usually been classified into two types according to changes in the polar vortex structure during the evolution of SSW events (Yoden et al., 1999; Ryoo and Chun, 2005; CP07). Yoden et al. (1999) used EOF analysis and the zonal wavenumber (ZWN) 1 and 2 amplitudes of the geopotential height at 58° N and 3 hPa at Lag = −5. Ryoo and Chun (2005) classified SSW events as ZWN-1 and -2, based on the relative amplitude of geopotential height perturbations at 65° N, 10 hPa, and Lag = ±7. CP07 classified all SSW events as a vortex displacement and vortex split using the absolute vorticity at 10 hPa. These previous studies did not include the SSW events that occurred after the mid-2000s and only used one or two reanalysis data sets. Thus, further statistical analysis of the differences between the two types of SSW events is needed.

As mentioned earlier, because large-scale planetary waves (PWs) have been considered to be important in the generation of SSW, the relationship between small-scale gravity waves (GWs) and SSW has not been thoroughly studied. In association with the development of GW observations using LIDAR or satellites, studies on GW variations during SSW have been performed recently. Using Rayleigh LIDAR, Duck et al. (1998) and Duck et al. (2001) showed that GW activity was reduced in the vortex core but enhanced along the vortex edge. In particular, they found that strong GW activity during December is associated with the warming in the vortex center. Using CHAMP/GPS satellite data, Wang and Alexander (2009) documented enhanced GW activity along the vortex edge in association with SSW. Recently, GW variation studies using parameterized GWD have been conducted. Limpasuvan et al. (2012) examined the contributions of PWs and GWs to the circulation and the thermal structure in the middle atmosphere using the 55-year WACCM3.5 model data and showed that westward GWD induced poleward and downward motion above 45 km altitude, resulting in adiabatic warming, an increase in the stratopause temperature, and vortex weakening in specific SSW event. More recently, Albers and Birner (2014), using the JRA-55 reanalysis data set, discovered that GWD was enhanced in the vortex edge during split SSW events, which changed the vortex structure to easily generate SSW through resonance. However, research on the contributions of PWs and GWs to residual mean circulation and temperature changes during the evolution of SSW events, which can be directly calculated using the downward control principle, has not yet been performed.

In this study, we examine the contributions of each forcing term in Transformed Eulerian mean (TEM) equations to the temperature changes during the evolution of SSWs that have been selected from four global reanalysis data sets during 34 years (1979–2012). The differences in the contributions between two types of SSW, Type-1 and Type-2, are also examined in details, through composite analyses in the residual mean circulation induced by the forcing terms and resultant temperature advection.

The present paper is structured as follows. In section 2, we explain the characteristics of the data sets and the methods of SSW selection and classification. In section 3, the wave forcing on stratospheric circulation and temperature change due to the wave forcing based on the downward control principle are described. Section 4 provides detailed results from the statistical analysis of SSW occurrence using four reanalysis data sets, contributions of PWs and GWs to the residual mean circulation, and contributions of each term in the TEM thermodynamic energy equation to temperature changes for the two types of SSW events using the MERRA data set. In addition, the same analysis using the ERA-Interim is shown in this section to evaluate the sensitivity among the data sets. Finally, we summarize our findings in section 5.

## 2. Data and SSW selection

Four reanalysis data sets are used: MERRA (Rienecker et al., 2011), ERA-Interim, NCEP-NCAR, and JRA-55 (Kobayashi et al., 2015). The characteristics of these reanalysis data sets are presented in Table 1. We used 34 years (from January 1979 to December 2012) of northern winter data and all variables are averaged over one day unless otherwise noted. An anomaly field is defined as the departure from the 34-year climatology. We used the following variables: zonal wind velocity (u), meridional wind velocity (v), vertical wind velocity (ω), temperature (T), and geopotential height (h). A parameterized zonal GWD is additionally used when we use the MERRA data set. This parameterized GWD is the sum of orographic GWD





(McFarlane, 1987) and non-orographic GWD (Garcia and Boville, 1994). This GWD is also used in the study of the variation of Brewer-Dobson circulation during the recent 30 years (Kim et al., 2014).

The SSW events are identified following CP07, using the four reanalysis data sets, and only SSW events that are present in all reanalyses are used in this study for robustness. All SSW events are classified as Type-1 or Type-2, based on

the criteria of Ryoo and Chun (2005) instead of using the more complicated method of CP07. In this study, central date of SSW (Lag = 0) is defined as the day when zonal-mean zonal wind at 60° N and 10 hPa is reversed to easterly. We define that N days before (after) the central date of SSW is denoted by Lag = +N (Lag = –N), then a SSW event for which the ZWN-1 amplitude of the geopotential height perturbation at 65° N, 10 hPa is larger than the ZWN-2 component at all times within Lag = ±7 is categorized as Type-1, while an SSW event for which the ZWN-2 component is larger than the ZWN-1

component at anytime within Lag = ±7 will be categorized as Type-2. The central dates and types of each SSW event are presented in Table 2. More detailed analyses are provided in section 4.1.

### 3. Methodology

#### 3.1 Residual mean circulation

The residual mean circulation in the middle atmosphere is represented in log-pressure coordinates as follows using residual

mean meridional ($\bar{v}^*$) and vertical velocities ($\bar{w}^*$) (Andrews et al., 1987).

$$\bar{v}^* = \bar{v} - \frac{1}{\rho_o}\frac{\partial}{\partial z}\left(\frac{\rho_o \overline{v'\theta'}}{\partial\bar{\theta}/\partial z}\right), \tag{1}$$

$$\bar{w}^* = \bar{w} + \frac{1}{a\cos\phi}\frac{\partial}{\partial\phi}\left(\cos\phi\,\frac{\overline{v'\theta'}}{\partial\bar{\theta}/\partial z}\right), \tag{2}$$

$$z = -H\ln\left(\frac{p}{p_s}\right), \tag{3}$$

$$\rho_o = \rho_s e^{-z/H} \approx \frac{p}{gH}. \tag{4}$$

Here, v and w are the meridional and vertical velocities, respectively. $\theta$ is potential temperature, and $\rho_o$ and $\rho_s$ are background air density and surface density, respectively. The term a is the radius of the Earth (a ≈ 6371 km), $\phi$ is latitude, $H$ is scale height ($H = 7$ km), p and $p_s$ are air pressure and surface pressure ($p_s = 1,000$ hPa), respectively. The bar symbol ($\bar{\phantom{x}}$) denotes a zonally averaged field, and perturbation ($'$) represents the departure from the zonal-mean field.

To examine the contributions of each forcing term to the residual mean circulation in Eqs. (1) and (2), the generalized

downward control principle in Eqs. (5) and (6) is used (Randel et al., 2002; Chun et al., 2011). It is worth noting that this 'generalized' (not considered stead-state) downward control principle should be used in the study of SSW because the zonal-mean zonal wind changes dramatically during the evolution of the SSW events.

$$\bar{v}^* = -\frac{1}{\rho_o\cos\phi}\frac{\partial}{\partial z}\left[-\cos\phi\int_z^\infty \rho_o\left(\frac{\text{EPD}+\overline{\text{GWD}}+\bar{X}-\frac{\partial\bar{u}}{\partial t}}{f_a}\right)dz'\right], \tag{5}$$

$$\bar{w}^* = \frac{1}{\rho_o a\cos\phi}\frac{\partial}{\partial\phi}\left[-\cos\phi\int_z^\infty \rho_o\left(\frac{\text{EPD}+\overline{\text{GWD}}+\bar{X}-\frac{\partial\bar{u}}{\partial t}}{f_a}\right)dz'\right], \tag{6}$$

$$f_a = f - \frac{1}{a\cos\phi}\frac{\partial}{\partial\phi}(\bar{u}\cos\phi), \tag{7}$$

where EPD and GWD are PW and GW forcing, respectively. As mentioned in section 2, we use the parameterized GWD provided from the MERRA data set. The term X is a residual term of the TEM momentum equation in Eq. (8), which contains the imbalance caused by the incremental analysis. These three terms operate as the forcing term in Eq. (8) to induce the mean wind change (Andrews et al., 1987). Eqs. (5) and (6) should originally be integrated along the same absolute

angular momentum lines; however, we use integration along constant latitude lines because the absolute angular momentum lines are almost parallel to the lines of constant latitude higher than 30° N (not shown).





### 3.2 TEM equations

To investigate the mean wind change induced by the wave forcing, we use the TEM momentum equation represented in Eq. (8) (Andrews et al. 1987).

$$\frac{\partial \bar{u}}{\partial t} - \bar{v}^* f_a + \bar{w}^* \frac{\partial \bar{u}}{\partial z} = EPD + \overline{GWD} + \bar{X}. \tag{8}$$

The TEM thermodynamic energy equation represented in Eq. (9) is also used to analyze the effect of each forcing term on the mean temperature changes.

$$\frac{\partial \bar{T}}{\partial t} + \frac{1}{a} \bar{v}^* \frac{\partial \bar{T}}{\partial \phi} + \bar{w}^* \left( \frac{\partial \bar{T}}{\partial z} + \kappa \frac{\bar{T}}{H} \right) = \bar{Q} - \frac{1}{\rho_o} \left( \frac{\partial B}{\partial z} + \kappa \frac{B}{H} \right) + \overline{X_T}, \tag{9}$$

$$B = \rho_o \left[ \frac{\overline{v'T'} \, \partial \bar{T}/\partial \phi}{a(\partial \bar{T}/\partial z + \kappa \bar{T}/H)} + \overline{w'T'} \right], \tag{10}$$

where $X_T$ is the residual term of the TEM thermodynamic energy equation, $\kappa$ is the gas constant (Rd = 287.05 J K$^{-1}$ kg$^{-1}$) divided by the specific heat of dry air at constant volume (Cp = 1005 J K$^{-1}$ kg$^{-1}$), and $Q$ is the diabatic heating rate, including atmospheric radiation, latent heating and moist processes, surface sensible heat flux, and turbulence effects. Radiation is the dominant source of diabatic heating rate in the northern hemisphere high-latitude regions in the stratsphere, which are the focus of this study (not shown).

The EPD in log-pressure coordinates is represented as follows (Andrews et al. 1987).

$$EPD = \frac{1}{\rho_o a \cos \phi} \nabla \cdot F = \frac{1}{\rho_o a \cos \phi} \left[ \frac{1}{a \cos \phi} \frac{\partial}{\partial \phi} \left( F^{(\phi)} \cos \phi \right) + \frac{\partial F^{(z)}}{\partial z} \right], \tag{11}$$

$$F^{(\phi)} = \rho_o a \cos \phi \left( \frac{\partial \bar{u}}{\partial z} \frac{\overline{v'\theta'}}{\partial \bar{\theta}/\partial z} - \overline{u'v'} \right), \tag{12}$$

$$F^{(z)} = \rho_o a \cos \phi \left( f_a \frac{\overline{v'\theta'}}{\partial \bar{\theta}/\partial z} - \overline{u'w'} \right). \tag{13}$$

### 4. Results

#### 4.1 Statistics of SSW events revealed in reanalysis data sets

Central dates and types of SSW events selected based on the criteria during a 34-year period are presented in Table 2. In total, 22, 23 and 24 SSW events are selected using the NCEP-NCAR, 2 reanalysis data sets (MERRA and ERA-Interim) and another data set (JRA-55), respectively. The NCEP-NCAR data set did not identify the March 1981 event as a MSW. The February 1981 event was identified as a MSW only in JRA-55. We use the 22 SSW events identified in all four reanalysis

data sets, and they were categorized as the same types. A total of 12 and 10 SSW events are categorized as Type-1 and Type-2, respectively. Hu et al. (2014) identified and categorized the SSW events in 55 years (1958–2012) of NCEP-NCAR data based on the criteria of CP07. Compared to Hu et al. (2014), four SSW events were categorized as different types (the Dec. 1998, Feb. 1999, Jan. 2006, and Feb. 2007 events).

    The number of SSW events that occurred during the 34 years are presented in Table 3. It can be seen that the number of

SSW events per winter is 0.676 (MERRA and ERA-Interim), 0.647 (NCEP-NCAR) and 0.706 (JRA-55), respectively. The ratio of Type-1 to Type-2 has a value between 1.2 and 1.4 in each reanalysis data set. Because more SSW events have occurred since 2002, the last year used in CP07, the number of SSW events per winter in CP07 (0.60 in the NCEP-NCAR and 0.64 in the ERA-40) is lower than our result. An increase in the frequency of SSW events over the last 30 years (1980–2009) was also shown in Kim et al. (2014).

Figure 1 is a time series of the zonal-mean zonal wind at 60° N and 10 hPa (upper panel of each event) and the amplitude of geopotential height perturbation at 65° N and 10 hPa (lower panel of each event) calculated using the MERRA reanalysis data. Although we used a total of four reanalysis data sets in the selection of SSW events, only the MERRA reanalysis data were used for analysis of circulation and temperature changes during the SSW events. The robustness of the





current result is examined by additional analysis using the ERA-Interim, which will be discussed in section 4. The ZWN-1 and -2 components of the geopotential height perturbation calculated through FFT are shown with the black and blue lines, respectively. The average values for Type-1 and Type-2 are represented in the third and fourth column of the last row. The temporal variations in the geopotential height perturbation during each SSW event are quite different, even in SSW events of

the same type. For instance, the December 1987 event was classified as Type-2 because of a rapid decrease in the ZWN-1 component. On the other hand, due to the strong enhancement of the ZWN-2 component nearly 15 days before Lag = 0, the January 2009 event was classified as Type-2. For Type-1 (composite mean), the ZWN-1 component of geopotential height perturbation is 2.5–3 times larger than the ZWN-2 component during the period of Lag = ±7. While the ZWN-1 component decreased gradually from Lag = –4 to Lag = +15, the ZWN-2 component barely changed from Lag = –15 to Lag = +15. For

Type-2 (composite mean), the ZWN-1 component is smaller and decreased earlier than that of Type-1. The ZWN-2 component is much larger than that of Type-1, but the ZWN-1 component is also larger than ZWN-2 in Type-2. This is because enhancement of the ZWN-2 component for Type-2 did not last long during most Type-2 SSW events.

We conducted two sensitivity tests of the SSW type categorizing criteria to evaluate the impact of the latitude and the period on classification of the SSW events. First, we obtained the same results when we used the geopotential height

perturbation at 60° N (same as the criteria of SSW occurrence using the zonal-mean zonal wind) as at 65° N. Second, if we used a shorter period (within Lag = ±4) to compare the ZWN-1 and -2 components, a total of 5 Type-2 events (Dec. 1987, Dec. 1998, Feb. 2001, Jan. 2003, and Feb. 2007) are recategorized as Type-1 events. On the other hand, if we use a longer period (within Lag = ±10), the results are same as the original ones. Thus, the SSW type categorizing criteria we used in this study is considered to reasonable, of which the results are also most similar to the results of Hu et al. (2004).

**4.2 Residual circulation induced by each forcing**

Figure 2 shows composited vertical profiles of residual mean (a) meridional and (b) vertical velocities for all 22 SSW events for the period Lag = –15 to Lag = +15 averaged over 70° N–80° N. Here, thick black lines denote residual mean meridional and vertical velocities defined by Eqs. (1) and (2), respectively. Thick blue lines signify residual mean velocities calculated using the total forcing term based on the generalized downward control principle, as noted in Eqs. (5) and (6). Thin lines

indicate residual mean velocities induced by EPD (green), GWD (red), the residual term (yellow), and zonal-mean zonal wind tendency (purple), respectively. It can be seen that the thick black lines are similar to the thick blue lines in the stratosphere. Thus, it can be assumed that understanding the residual mean circulation using each forcing term of the TEM equation is reasonable during the evolution of SSW events. It should be noted that some differences exist at an altitude greater than 1 hPa due to the lack of layers used for integration in Eqs. (5) and (6); hence, we used data only up to 1 hPa

(~48.35 km).

EPD is the dominant component of the total residual mean circulation. EPD induced enhanced poleward motion and strong downward motion during the SSW evolution. GWD usually induced residual mean circulation in the same direction as the circulation induced by EPD. Though its amplitude is much smaller than that induced by EPD, the contribution of GWD to the vertical component ($\overline{w}^*$) in the upper stratosphere is considerable. The effect of the residual term on the residual

mean circulation is much weaker than that of other wave forcing. The zonal-mean zonal wind tendency term produced opposite circulation (equatorward and upward motion) as the circulation induced by EPD. It was found that the circulation produced by the wind tendency is maximal at approximately 10 hPa, where the stratospheric temperature mainly increases during SSW events. Hence, the circulation partially cancelled the adiabatic warming effects generated by EPD and GWD.

Figure 3 shows a time-height cross section of EPD, GWD, residual term, and zonal-mean zonal wind tendency (shading)

and the residual mean circulation (red arrow) induced by each forcing term. While all forcings are averaged over 60° N–70° N, the residual mean circulation is averaged over 70° N–80° N. A negative (positive) forcing causes clockwise (counter clockwise) circulation in the northern hemisphere and eventually leads to diabatic warming (cooling) in the polar





stratosphere. The red arrow in the bottom right corner of each figure denotes the reference vector, which has a magnitude of 5 m s$^{-1}$ (meridional velocity) and 1.67 mm s$^{-1}$ (vertical velocity). The reference vectors in the third (GWD) and fourth (residual term) columns are exaggerated by three times for better presentation. The positive x and y directions are poleward and upward, respectively. We used daily averaged values for each forcing term and residual mean circulation and present the arrows once for every two days for easy visualization.

Figures 3a and 3c show composites of each forcing and residual mean circulation for Type-1 and Type-2, respectively. The first column represents the total forcing and circulation based on the downward control principle. It was found that a strong negative forcing exists in the stratosphere during Lag = −5 to Lag = 0 for both types of SSW events, although Type-2 has larger values than Type-1. The second, third, fourth, and last columns represent the residual mean circulation induced by the EPD, GWD, residual term, and zonal-mean zonal wind tendency, respectively. Here, the wind tendency term includes a minus sign, as in Eqs. (5) and (6). EPD is the main contributing factor to the residual mean circulation among the forcing terms. A negative EPD in Type-2, which has a larger magnitude and exists at a lower altitude than that in Type-1, produces a poleward and downward motion just before the central date of SSW events. The wind tendency partially cancels the effect of EPD by inducing residual mean circulation in the opposite direction. Similar to the EPD, the wind tendency in Type-2 has a larger magnitude than that in Type-1.

Although GWD forcing mostly features small negative values, a large proportion of the total downward motion in the upper stratosphere can be induced by it. As the time approaches Lag = 0, the magnitude of the negative GWD forcing gradually decreases. After Lag = 0, there is a forcing sign reversal (negative to positive) in the upper stratosphere. This is because the GWs that have zero phase speed are filtered out in the stratosphere as the zonal-mean zonal wind changes from westerly to easterly during the evolution of SSW events. Under this situation, eastward propagating (relative to the mean wind) GWs deposit the positive momentum in the upper stratosphere, as shown in Fig. 3a and 3c. Note that the GWD forcing provided in the MERRA data set includes not only the orographic GWs (which have zero phase speed) but also the non-orographic GWs (which have a various phase speed range). The GWs associated with jet/frontal system (Sato et al. 2009; Kim et al. 2016) and convective clouds (Ern et al. 2011) can be candidates of these waves, and the increase in positive GW momentum flux up to Lag = 0 implies for the existence of eastward-propagating GWs and westward-propagating GWs with phase speeds less than the background wind. It would be necessary to conduct further research to determine the contributions of the orographic and non-orographic GWs to the circulation during SSW events using an atmospheric model simulation. The residual term has a much smaller magnitude than other forcing terms.

To reveal the net effects of each forcing on the residual mean circulation during the evolution of SSW, the climatological signal should be removed from the original fields. Figures 3b and 3d are the anomaly fields of each forcing term and circulation shown in Fig. 3a and 3c, respectively. Figure 3e shows the difference between Fig. 3d (Type-2 anomaly) and 3b (Type-1 anomaly). The hatch patterns denote statistical significance at the 95% confidence level. The negative EPD forcing is enhanced by more than 20 m s$^{-1}$ day$^{-1}$ for both types at approximately Lag = 0. However, the EPD forcing anomaly of Type-1 is greatest during the period between Lag = −3 and Lag = −1 and has a secondary peak at approximately Lag = −8. In contrast, a much stronger single peak exists at Lag = −3 for Type-2. These negative EPD anomalies induce enhanced poleward and downward motion during the periods with large EPD terms. Differences in the EPD anomalies between the two types of SSW events are evident at Lag = −5 to −1 at 30–40 km altitude. After the central date of a SSW event, the difference is significant at Lag = +9 due to large anomalous positive EPD values in Type-2. The wind tendency anomaly fields are quite similar to the original fields, as statistically significant differences exist in the upper stratosphere at Lag = −5 to −2. The GWD forcing anomaly has different structures in the two types of SSW events. In the period between Lag = −15 and −5, the GWD anomaly in Type-1 has small positive and negative values, while negative GWD forcing is enhanced in Type-2 mostly in the upper stratosphere. The largest difference between the two GWD anomalies exist at





approximately Lag = −15. This is consistent with the result that GWD forcing is enhanced in association with the polar night jet for the vortex split type SSW reported by Albers and Birner (2014).

To ensure the difference in the GWD for the two types of SSW, polar-stereo projection maps of GWD anomalies averaged over the upper stratosphere (10–1 hPa) at Lag = −15 are shown in Fig. 4a. Latitude circles are drawn every 10°

from 50° N and meridians every 30°. At 60° N–70° N, where the GWD anomalies are averaged in Fig. 3, a positive (negative) GWD anomaly mainly exists for Type-1 (Type-2) SSW events. These GWD anomalies with different signs are strong in mountain regions (e.g., the Rocky Mountains in Canada, Scandinavian Mountains, and the Verkhoyansk range in eastern Siberia). Figure 4b shows vertical profiles of the zonal-mean zonal wind averaged over 60° N–70° N at Lag = −15. The black line and blue line indicate the zonal-mean zonal wind and its anomaly, respectively. For Type-1 SSW events,

negative zonal-mean zonal wind anomalies are dominant due to the weakening of westerlies in the stratosphere. Thus, GWs that have positive phase speed can easily propagate and break into the upper stratosphere, resulting in anomalous positive GWD values in that region. However, for Type-2 SSW events, more eastward propagating GWs are filtered out due to slightly enhanced westerlies; thus, negative GWD forcing is enhanced in the upper stratosphere. Although the magnitude of the residual mean circulation induced by the GWD forcing is much smaller than that by the EPD during SSW events, it is

worth noting that an indirect effect of GWD, i.e., enhanced GWD forcing can change the geopotential field and finally change the PW forcing, should be investigated to further understand the contribution of GWs to SSW, which remains for future research. The residual term anomaly has a similar structure as the original field with a smaller magnitude than the other forcing terms.

### 4.3 Mean temperature change by each forcing term

Time-height cross sections of the zonal-mean temperature anomaly in the polar stratosphere are shown in Fig. 5a. The left, middle, and right columns represent Type-1, Type-2, and the difference between Type-2 and Type-1, respectively. Clear sudden temperature increases are present in both types: at Lag = −7 to −6 and Lag = 0, for Type-1 and at Lag = 0 for Type-2, which exhibits greater warming. The maximum temperature anomaly is approximately 16–18 K for Type-1 and 20–22 K for Type-2. A significant difference in the temperature anomalies is highlighted in two regions: in the height range of 30–40 km

at Lag = −7 to −6 and 20–30 km at approximately the central date. In contrast to the warming pattern in the middle stratosphere, a cooling pattern is dominant in the upper stratosphere and lower mesosphere, with a gradual decreasing signal to the lower atmosphere.

Figure 5b is a time-height cross section of the zonal-mean temperature tendency anomaly averaged over 70° N–80° N. The structure of the temperature anomaly in Fig. 5a is affected by the structure of the prior temperature tendency anomaly

shown in Fig. 5b. In other words, a large temperature tendency at approximately Lag = −9 and Lag = −2 to −1 produces temperature anomaly maxima at Lag = −7 to −6 and at approximately Lag = 0, respectively, with magnitudes of approximately 2 K day$^{-1}$ for Type-1. Similarly, for Type-2, a stronger temperature tendency anomaly at Lag = −3 induces a temperature anomaly maximum at approximately Lag = 0 with a magnitude of approximately 4 K day$^{-1}$. A statistically significant difference between the two types of temperature tendency anomalies exists at Lag = −5 to −2. While the warming

signal lasts until 2 weeks after Lag = 0 (see Fig. 5a), the sign of the temperature tendency anomaly changes from positive (before Lag = 0) to negative (after Lag = 0). Thus, the temperature recovery phase starts at approximately Lag = 0. However, differences in the temperature recovery between the two types are not significant.

Based on Eq. (9), the TEM thermodynamic energy equation consists of a zonal-mean temperature tendency, a temperature advection term associated with residual mean circulation, a diabatic heating rate, a term related to the eddy heat

flux, and a residual term for the TEM thermodynamic energy equation. Hereafter, we will call the fourth and fifth terms Eddy and Res-T, respectively. Figure 6 shows a time-height cross section for each term in the TEM thermodynamic energy equation: the zonal-mean temperature tendency (first column), the temperature advection (second column), the diabatic


heating rate (third column), the Eddy (fourth column), and the Res-T (fifth column) averaged over 70–80° N for both types of SSW. Only the anomaly fields are represented in Fig. 6 to show the net effects during the evolution of SSW events.

Just before the central date of SSW, a structure of the temperature advection anomaly is similar to that of the temperature tendency anomaly. The maximum temperature advection anomaly exists at a slightly higher altitude than the

level where the temperature tendency anomaly has a maximum value, with a larger magnitude. A negative diabatic heating rate (cooling) anomaly partially cancels the warming effect by the temperature advection anomaly. The difference in the adiabatic heating rates of the two SSW types is not statistically significant. Just after the central date of SSW, a negative temperature advection anomaly contributes to the temperature recovery in both types of SSW. Anomalously negative diabatic heating rate also contribute to the recovery of temperature in the stratosphere. During the whole period, the effect of

the Eddy term is small compared to the two aforementioned terms. The magnitude of the Eddy term is within 2 K day$^{-1}$ for both types. Res-T is also much smaller in magnitude than the temperature advection and the diabatic heating rate. In summary, temperature changes are mainly induced by the temperature advection before Lag = 0 and by the temperature advection and the adiabatic heating rate after Lag = 0. Therefore, determining the contribution of each forcing to temperature advection (which can produce strong warming before the central date of SSW events) will be an important next step to

understating the mechanism responsible for SSW.

Figure 7 shows the temperature advection by the residual mean circulation led by each forcing term. Each column has the same meaning as in Fig. 3 except the temperature advection is averaged over 70° N–80° N. Only the anomaly fields are represented in Fig. 7 to show the net effects during the SSW period. The overall pattern of the total temperature advection anomalies is quite similar to that of the temperature advection anomalies by EPD, although the magnitude of the latter is

larger. In other words, the total temperature advection anomalies are induced primarily by the EPD for both types of SSW, with some cancellation by other terms. Wind tendency anomalies generally have opposite effects on temperature advection as those of EPD. Quantitatively, the temperature advection anomaly maximum (minimum) induced by EPD (wind tendency) is approximately 10 K day$^{-1}$ (–4 K day$^{-1}$) at Lag = –1 for Type-1. On the other hand, for Type-2, the maximum (minimum) magnitude is approximately 13 K day$^{-1}$ (–5 K day$^{-1}$) at Lag = –3. In the upper stratosphere, GWD usually results in small

negative temperature advection anomalies for the whole period for Type-1, whereas weak positive temperature advection anomalies are induced for the period from Lag = –15 to Lag = –7 for Type-2, because the enhanced negative GWD produces poleward and downward motion that can generate adiabatic warming in the polar stratosphere. However, statistically significant differences do not exist in the GWD temperature advection between the two types of SSW. The magnitude of the temperature advection anomalies induced by the residual term is less than or similar to 1 K day$^{-1}$.

To examine the latitude-height structure of each term in the TEM thermodynamic energy equation before and after the central date of SSW events, latitude-height cross sections of the zonal-mean temperature tendency, temperature advection, and diabatic heating rate at Lag = –2 (Lag = +2) are shown in Fig. 8 (Fig. 9). Eddy and Res-T (not shown) are excluded in these figures because they have relatively small magnitudes compared to the other three terms. All figures are represented as anomalies with respect to the climatology to emphasize the effect of each term during the SSW. In the second column, the

blue arrow denotes the residual mean circulation vector $(\bar{v}^*, \bar{w}^*)$, which is averaged over every 4 grid points (6 degrees) for clear visualization.

Figure 8 shows the temperature change structure at Lag = –2 when strong warming exists in both types of SSW. The temperature tendency anomalies have positive values poleward of 60° N that gradually increase with increasing latitude. For Type-1, the temperature tendency anomaly has a maximum value of 3 K day$^{-1}$ at 30–40 km altitude. For Type-2, the

maximum temperature tendency anomaly is at 20–35 km altitude and has a value of 5 K day$^{-1}$. For both types of SSW, positive temperature advection anomalies extend from the upper stratosphere in the middle latitudes to the whole stratosphere in the polar region. Their patterns are quite similar to that of temperature advection; that is, at this time, strong stratospheric warming events are primarily induced by temperature advection. Significant differences between the two





temperature tendency anomalies exist at 20–30 km altitude due to the stronger positive temperature advection for Type-2 than Type-1. The diabatic heating rates have anomalously negative values (radiative cooling) with minima (less than $-1$ K day$^{-1}$ for Type-1 and $-2$ K day$^{-1}$ for Type-2) at 30–45 km altitude. This anomalous cooling partially cancels the warming induced by the temperature advection.

A minimum zonal-mean temperature tendency anomaly exists at approximately Lag = +2 (See Fig. 5b), although temperature anomaly itself continues to increase (See Fig. 5a) at this time. Therefore, it is expected that the strong temperature recovery exists at Lag = +2. Figure 9 is same as Fig. 8, except at Lag = +2. It can be seen that an anomalously negative temperature tendency (up to $-4$ K day$^{-1}$ and $-2$ K day$^{-1}$ for Type-1 and Type-2, respectively) exists in the polar upper stratosphere. There are two reasons why negative temperature tendency anomalies exist at this time. First, the

temperature advection anomalies changed in sign from positive to negative at approximately Lag = 0. Additionally, anomalously negative temperature advection exists above 20 km and above 30 km for Type-1 and Type-2, respectively. Second, the anomalously negative diabatic heating rate maintains its sign until few days after the central date of the SSW (See fig. 6). At Lag = +2, anomalous cooling with values of less then $-1$ K day$^{-1}$ is induced by diabatic forcing, especially below 40 km altitude. As a result, these diabatic cooling effects extended the negative temperature tendency anomaly to

lower altitude.

To summarize, warming patterns in the polar stratosphere just before the central date of the SSW were induced primarily by temperature advection (adiabatic heating) by EPD, while anomalous temperature cooling just after Lag = 0 occurred due to reduced temperature advection and anomalous diabatic cooling effects.

To examine the robustness of the reanalysis data, the same analyses were performed using the ERA-Interim reanalysis

data set. We used the native model-level (up to 0.1 hPa) data of ERA-Interim because the lack of layers in the conventional pressure level data (up to 1 hPa) in the upper stratosphere can produce some errors when we use the downward control principle (Okamoto et al., 2011). Figure 10 shows the time-height cross sections of zonal-mean temperature tendency, total temperature advection, and temperature advection induced by the EPD and wind tendency calculated using MERRA (Fig. 10a and c) and ERA-Interim (Fig. 10b and d). All values are anomaly fields. The patterns produced using the MERRA are

ERA-Interim data sets are very similar; that is, warming (cooling) patterns just before (after) the Lag = 0 are clear, and the opposite effects of the EPD and the wind tendency on temperature advection are also confirmed. Therefore, the results of this paper are not specific to just one data set but are common to multiple data sets.

### 5. Summary

A composite analysis for total 22 SSW events from 1979 to 2012 (34 years) was performed to investigate the residual mean

circulation and temperature changes during the evolution of the SSW events using four reanalysis data sets. The SSW events are classified as Type-1 (12 events) and Type-2 (10 events) based on the relative amplitudes of ZWN-1 and -2 at 65° N and 10 hPa. This classification method is simpler than the criteria of CP07, and the results are quite similar. The frequency of SSW events and the ratio of Type-1 to Type-2 do not reveal noticeable differences among the reanalysis data sets. Due to the frequent occurrence of SSW events in the late 2000s, a higher frequency of SSW events during winter exist compared to the

study of CP07, which used the period between from 1958 to 2002. The residual mean circulation induced by each forcing term in the TEM momentum equation is calculated for both types of SSW based on the generalized downward control principle using the MERRA reanalysis data set. Strong poleward and downward motion exists at approximately Lag = $-8$ and Lag = $-1$ for Type-1 and at Lag = $-3$ for Type-2. This motion is produced primarily by the EPD and partially cancelled by the wind tendency. The residual mean circulation is stronger for Type-2 than Type-1. At approximately Lag = $-14$, a

negative GWD is weakened for Type-1 but enhanced for Type-2; these differences are caused by differences in the GW filtering at that time. Statistically significant temperature and temperature tendency differences exist between the two types





of SSW. It can be seen that the temperature tendency patterns are similar to those of the temperature advection before the central date of SSW events and that diabatic cooling partially cancels the warming effect.

EPD is the most significant contribution to the temperature advection, whereas the contribution of GWD is relatively small. At Lag = –2, the period when strong warming exists in the polar stratosphere for both types of SSW, the temperature tendency anomalies are stronger in Type-2 than in Type-1 because the downward motion primarily induced by the EPD is stronger in Type-2 than in Type-1 at this time. Anomalous diabatic cooling partially cancels the warming effect produced by temperature advection in the upper stratosphere. On the other hand, at Lag = +2, the temperature tendency anomalies have negative values due to the negative temperature advection anomalies and diabatic cooling. The same analysis was conducted using the ERA-Interim reanalysis data set to examine the dependency of the data sets. Very similar patterns of temperature tendency and temperature advection were observed during the evolution of the SSW events.

*Acknowledgements.* The MERRA data set was obtained from http://disc.sci.gsfc.nasa.gov/daac-bin/DataHoldings.pl. The ERA-Interim data set is available through http://apps.ecmwf.int/datasets/data/interim-full-daily/levtype=pl. The NCEP-NCAR data set was from http://www.esrl.noaa.gov/psd/data/gridded/data.ncep.reanalysis.html. The JRA-55 data set was provided from http://rda.ucar.edu/datasets/ds628.0. This work was supported by the Korea Polar Research Institute grant PE16090.

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

30

35

40



**Table 1. Characteristics of the reanalysis data sets. Here, u, v, ω, T, and h are the zonal, meridional, vertical velocity, temperature and geopotential height, respectively. MERRA provided the parameterized gravity wave drag (GWD) as explained in the text.**

| Data set | | MERRA | ERA-Interim | NCEP-NCAR | JRA-55 |
|---|---|---|---|---|---|
| Resolution | Temporal | 3 hours | 6 hours | 6 hours | 6 hours |
| | Horizontal | 1.25° x 1.25° | 1.5° x 1.5° | 2.5° x 2.5° | 1.25° x 1.25° |
| | Vertical | 42 levels | 37 levels | 17 levels | 37 levels |
| | (Top level) | (0.1 mb) | (1 mb) | (10 mb) | (1 mb) |
| Variables used in the present study | | u, v, ω, T, h, GWD | u, v, ω, T, h | | |



**Table 2. SSW events selected using four reanalysis data sets spanning 34 years (Jan. 1979–Dec. 2012). The SSW selection and categorization methods are described in the text.**

| MERRA | | | | ERA-Interim | | | | NCEP-NCAR | | | | JRA-55 | | | |
|---|---|---|---|---|---|---|---|---|---|---|---|---|---|---|---|
| Year | Mon | Day | Type | Year | Mon | Day | Type | Year | Mon | Day | Type | Year | Mon | Day | Type |
| 1979 | 2 | 22 | 2 | 1979 | 2 | 22 | 2 | 1979 | 2 | 22 | 2 | 1979 | 2 | 22 | 2 |
| 1980 | 2 | 29 | 1 | 1980 | 2 | 29 | 1 | 1980 | 3 | 1 | 1 | 1980 | 2 | 29 | 1 |
|  |  |  |  |  |  |  |  |  |  |  |  | 1981 | 2 | 6 | 1 |
| 1981 | 3 | 4 | 1 | 1981 | 3 | 4 | 1 |  |  |  |  | 1981 | 3 | 4 | 1 |
| 1981 | 12 | 4 | 1 | 1981 | 12 | 4 | 1 | 1981 | 12 | 4 | 1 | 1981 | 12 | 4 | 1 |
| 1984 | 2 | 24 | 1 | 1984 | 2 | 24 | 1 | 1984 | 2 | 24 | 1 | 1984 | 2 | 24 | 1 |
| 1985 | 1 | 1 | 2 | 1985 | 1 | 1 | 2 | 1985 | 1 | 2 | 2 | 1985 | 1 | 1 | 2 |
| 1987 | 1 | 23 | 1 | 1987 | 1 | 23 | 1 | 1987 | 1 | 23 | 1 | 1987 | 1 | 23 | 1 |
| 1987 | 12 | 8 | 2 | 1987 | 12 | 8 | 2 | 1987 | 12 | 8 | 2 | 1987 | 12 | 8 | 2 |
| 1988 | 3 | 14 | 2 | 1988 | 3 | 14 | 2 | 1988 | 3 | 15 | 2 | 1988 | 3 | 14 | 2 |
| 1989 | 2 | 21 | 2 | 1989 | 2 | 21 | 2 | 1989 | 2 | 22 | 2 | 1989 | 2 | 21 | 2 |
| 1998 | 12 | 15 | 2 | 1998 | 12 | 15 | 2 | 1998 | 12 | 15 | 2 | 1998 | 12 | 15 | 2 |
| 1999 | 2 | 26 | 1 | 1999 | 2 | 26 | 1 | 1999 | 2 | 26 | 1 | 1999 | 2 | 26 | 1 |
| 2000 | 3 | 20 | 1 | 2000 | 3 | 20 | 1 | 2000 | 3 | 21 | 1 | 2000 | 3 | 20 | 1 |
| 2001 | 2 | 11 | 2 | 2001 | 2 | 11 | 2 | 2001 | 2 | 12 | 2 | 2001 | 2 | 11 | 2 |
| 2001 | 12 | 30 | 1 | 2001 | 12 | 30 | 1 | 2002 | 1 | 1 | 1 | 2001 | 12 | 31 | 1 |
| 2003 | 1 | 18 | 2 | 2003 | 1 | 18 | 2 | 2003 | 1 | 18 | 2 | 2003 | 1 | 18 | 2 |
| 2004 | 1 | 5 | 1 | 2004 | 1 | 5 | 1 | 2004 | 1 | 6 | 1 | 2004 | 1 | 5 | 1 |
| 2006 | 1 | 21 | 1 | 2006 | 1 | 21 | 1 | 2006 | 1 | 21 | 1 | 2006 | 1 | 21 | 1 |
| 2007 | 2 | 24 | 2 | 2007 | 2 | 24 | 2 | 2007 | 2 | 24 | 2 | 2007 | 2 | 24 | 2 |
| 2008 | 2 | 22 | 1 | 2008 | 2 | 22 | 1 | 2008 | 2 | 22 | 1 | 2008 | 2 | 22 | 1 |
| 2009 | 1 | 24 | 2 | 2009 | 1 | 24 | 2 | 2009 | 1 | 24 | 2 | 2009 | 1 | 24 | 2 |
| 2010 | 2 | 9 | 1 | 2010 | 2 | 9 | 1 | 2010 | 2 | 9 | 1 | 2010 | 2 | 9 | 1 |
| 2010 | 3 | 24 | 1 | 2010 | 3 | 24 | 1 | 2010 | 3 | 24 | 1 | 2010 | 3 | 24 | 1 |





**Table 3.** Statistics of the SSW events selected from the different reanalysis data sets. The second, third, and fourth columns show the number of total SSW events, Type-1 SSW events, and Type-2 SSW events, respectively. The fifth column shows the frequency

5  of SSW events during the winter (November–March). The last column shows the ratio of Type-1 to Type-2 SSW events.

| Reanalysis | Total SSWs | Type-1 SSWs | Type-2 SSWs | SSWs/winter | Ratio |
|---|---|---|---|---|---|
| MERRA | 23 | 13 | 10 | 0.676 | 1.3 |
| ERA-Interim | 23 | 13 | 10 | 0.676 | 1.3 |
| NCEP-NCAR | 22 | 12 | 10 | 0.647 | 1.2 |
| JRA-55 | 24 | 14 | 10 | 0.706 | 1.4 |



**Figure 1. Time series of the zonal-mean zonal wind at 60° N and 10 hPa and the amplitude of the geopotential height perturbation of zonal wavenumber 1 (black) and 2 (blue) at 65° N and 10 hPa, calculated using the MERRA reanalysis data. Red (green) dashed lines denote Lag = 0 (Lag = −7 and Lag = +7). The last two figures in these panels show the composite results for the Type-1 and**

5  **Type-2 SSW events. The selection method of each type of SSW is included in the text.**



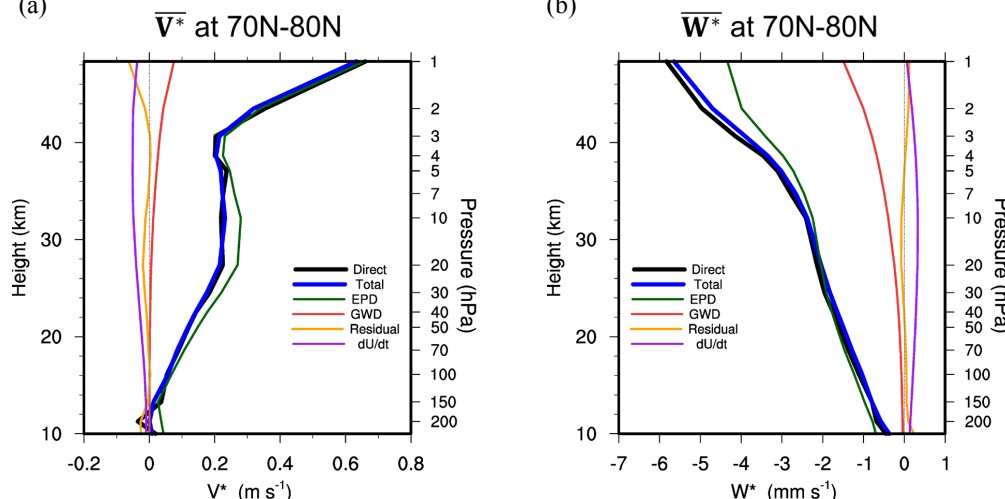

**Figure 2. Composited vertical profiles of residual mean (a) meridional and (b) vertical velocities throughout all SSW events for the period Lag = −15–+15 averaged over 70° N–80° N using the MERRA reanalysis data. Thick black lines denote circulations defined**

5 **by (1) and (2). Thick blue lines indicate values calculated using the total forcing term based on the downward control principle, as noted (5) and (6). Green, red, yellow, and purple thin lines indicate residual circulation induced by EPD, GWD, residual term, and zonal-mean zonal wind tendency, respectively.**







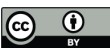



Figure 3. (a) Time-height cross sections of each forcing term averaged over 60° N–70° N (shading) and the residual mean velocities (red arrow) terms and total forcing term for Type-1 SSW events averaged over 70° N–80° N, induced by each forcing. (b) The same as in (a), except shown as anomalies from the 34-year climatological mean of each day. (c) and (d) are the same as (a) and (b), respectively, except for Type-2 SSW events. The difference between the anomaly fields of Type-2 and Type-1 SSW events is shown in (e). The hatch patterns in (e) denote statistical significance at 95% confidence level.





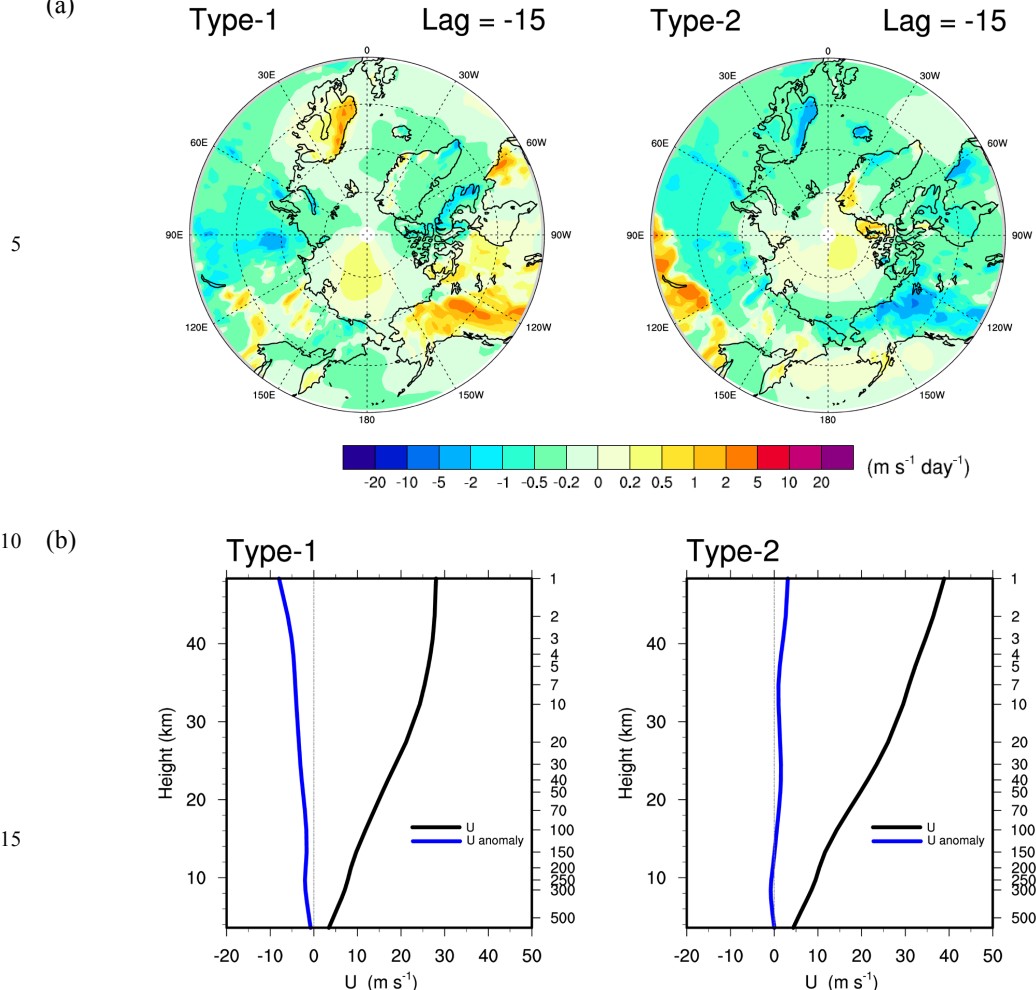

**Figure 4. (a) Polar-stereo projection maps of the GWD anomaly averaged in height between 10 and 1 hPa for Type-1 (left) and Type-2 (right) SSW events. (b) Vertical profiles of zonal-mean zonal wind averaged over 60° N–70° N in the 15 days before the central date.**



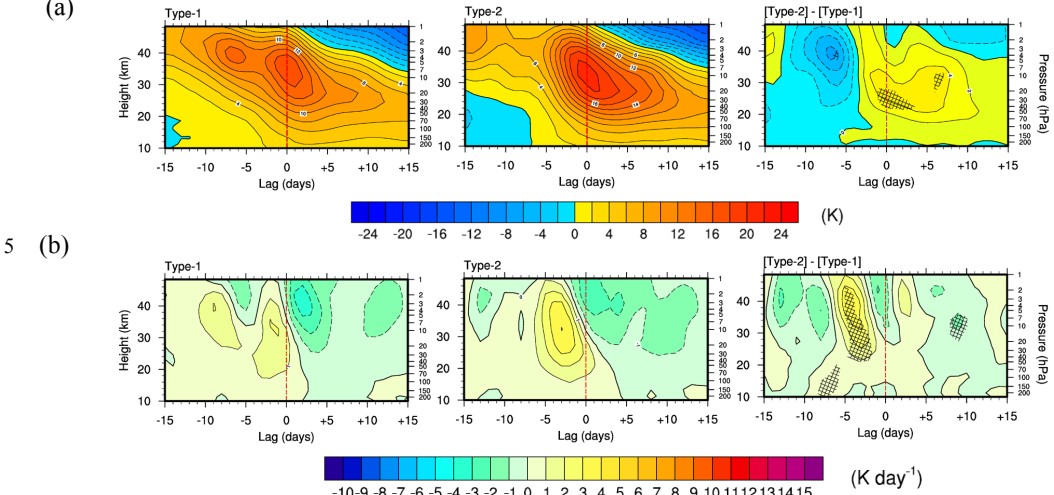

**Figure 5. (a)** Time-height cross sections of the temperature anomaly averaged over 70° N–80° N for Type-1 (left column), Type-2 (middle column), and the difference between (b) and (a) (right column). **(b)** Time-height cross sections of the temperature tendency anomaly averaged over 70° N–80° N for both types of SSW and the difference between the two types. The hatch patterns denote statistically significant differences in the temperature (temperature tendency) anomaly between Type-2 and Type-1 SSW events above a 90% (95%) confidence level.





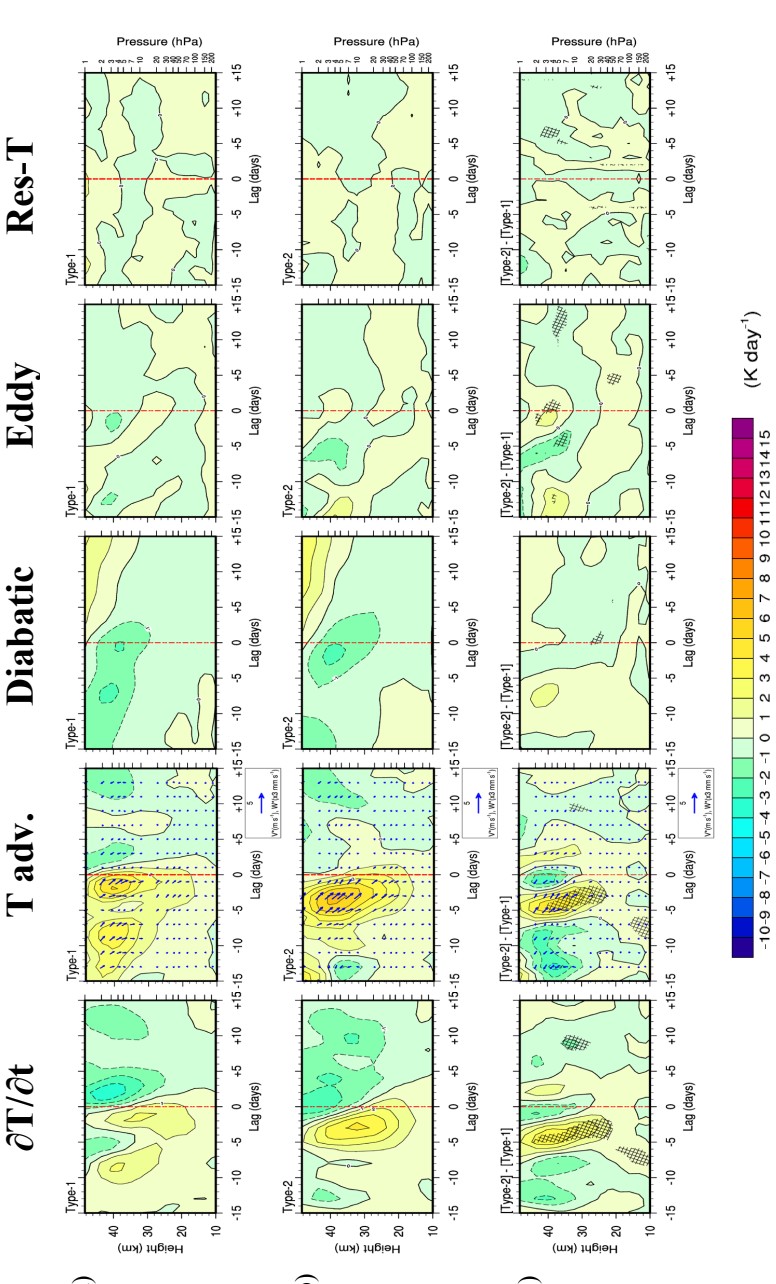

**Figure 6.** (a) Time-height cross sections of the zonal-mean temperature tendency (first column), temperature advection (second column), diabatic heating rate (third column), eddy heat flux (fourth column), and the residual term of TEM thermodynamic energy equation (fifth column) for Type-1 SSW events averaged over 70° N–80° N. All values are anomaly fields from the climatology. (b) is the same as (a), except for Type-2 SSW events. The difference between (b) and (a) is shown in (c).





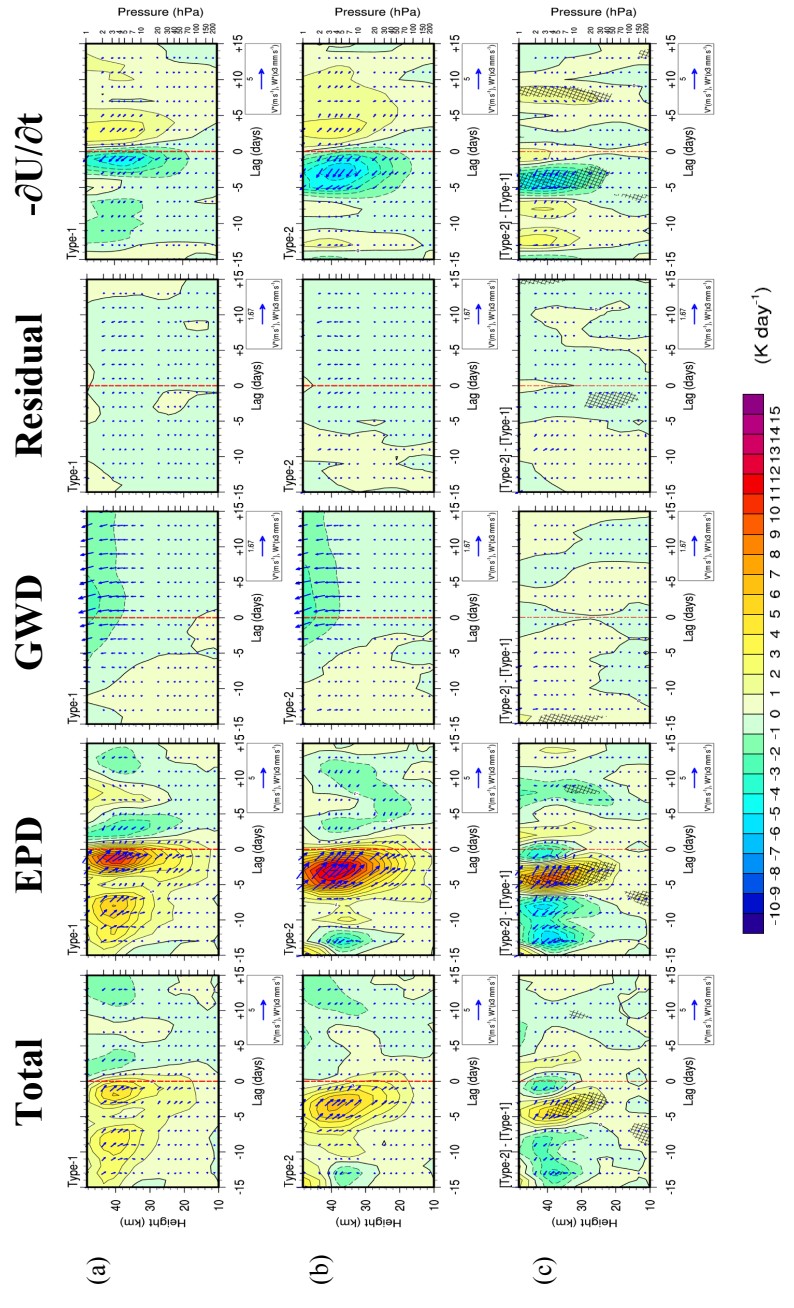

**Figure 7. The same as in Fig. 6, except for the temperature advection by residual mean circulation averaged over 70° N–80° N.**



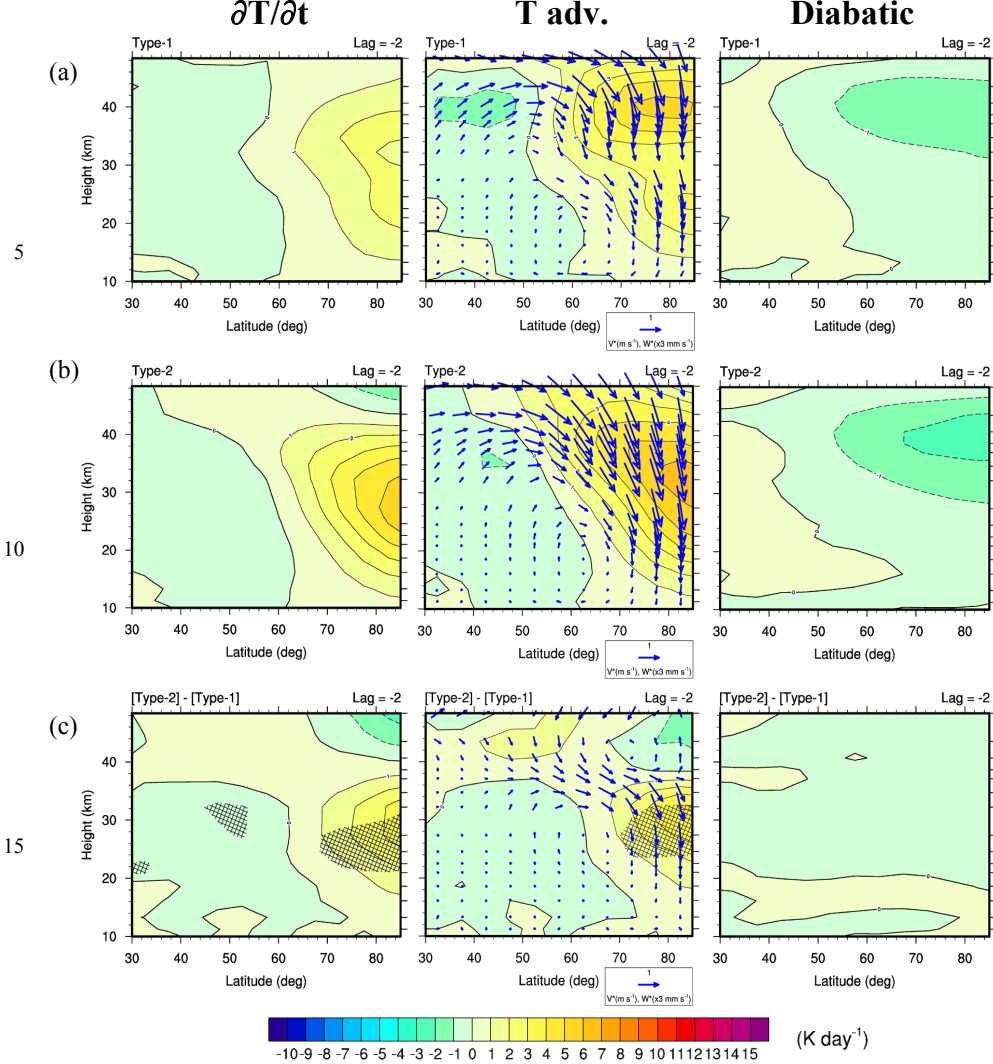

**Figure 8. (a) Latitude-height cross sections of the zonal-mean temperature tendency (left column), temperature advection (middle column), and diabatic heating rate (right column) at Lag = −2 for Type-1 SSW events. All values are anomaly fields from the climatology. (b) The same as (a), except for Type-2 SSW events. (c) The difference between (b) and (a). The blue arrow denotes the residual mean circulation induced by the total forcing.**





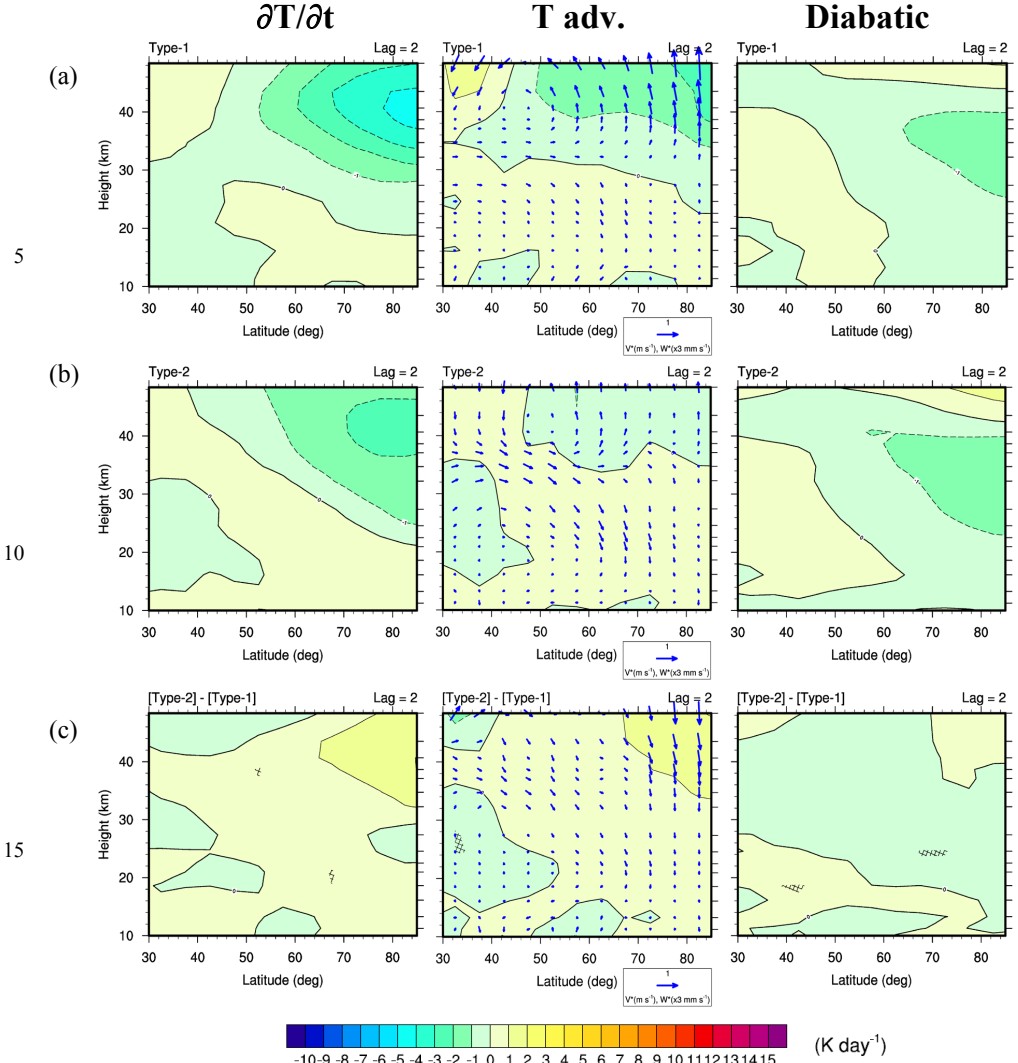

20    **Figure 9. The same as in Fig. 8, except at Lag = +2 for both types of SSW.**



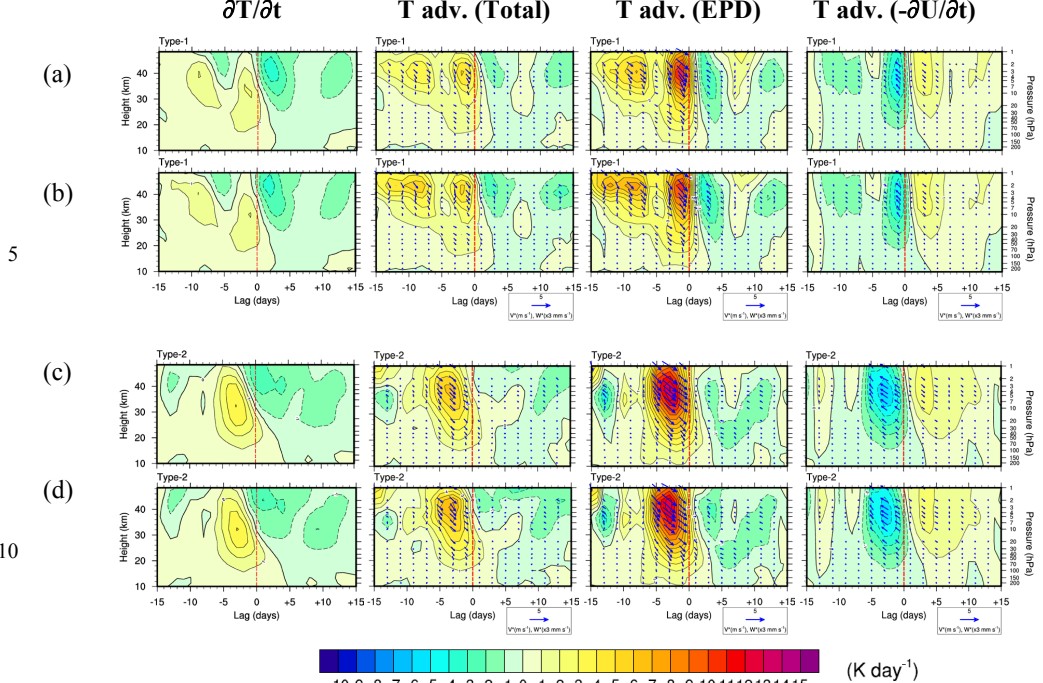

**Figure 10. (a)** Time-height cross sections of the zonal-mean temperature tendency anomaly (first column), total temperature advection anomaly (second column), temperature advection anomaly by EPD (third column), and temperature advection anomaly by wind tendency (fourth column) for Type-1 SSW events averaged over 70° N–80° N using MERRA. **(b)** The same as (a), except for using the model-level of ERA-Interim. **(c)** and **(d)** are the same as (a) and (b), respectively, except for Type-2 SSW events.