# Peer review of "Residual Mean Circulation and Temperature Changes during the Evolution of Stratospheric Sudden Warming Revealed in MERRA"

_Atmospheric Chemistry and Physics, 2016_

## Short Comment (SC1) · 6 Dec 2016

Dear authors,

I would like to ask you to extend your analysis range in Fig. 4a to approx. 30°N to capture the possible GWD anomalies "above Japan" (see the EA/NP hotspot in Šácha et al. (2015) and its possible implications in Šácha et al. (2016)). However, this hotspot is dominant in the vertical range between 30hPa and 10hPa, which is below your analysis range. Please consider as well to supplement your existing analysis (10hPa-1hPa, where the Scandinavia hotspot is dominant) with two additional plots of

GWD anomalies between e.g. 30hPa and 10hPa.

Best wishes, Petr Šácha.

Šácha, P., Kuchař, A., Jacobi, C., and Pišoft, P.: Enhanced internal gravity wave activity and breaking over the northeastern Pacific–eastern Asian region, Atmos. Chem. Phys., 15, 13097-13112, doi:10.5194/acp-15-13097-2015, 2015.

Šácha, P., Lilienthal, F., Jacobi, C., and Pišoft, P.: Influence of the spatial distribution of gravity wave activity on the middle atmospheric circulation and transport, Atmos. Chem. Phys. Discuss., doi:10.5194/acp-2016-548, in review, 2016.
* * *

---

## Referee Comment (RC1) · Anonymous Referee #1 · 9 Dec 2016

Referee report on "Residual Mean Circulation and Temperature Changes during the Evolution of Stratospheric Sudden Warming Revealed in MERRA"

by B.-G. Song and H.-Y Chun

This paper examines SSW in several reanalysis datasets and reaches two main conclusions:

1. That the various datasets yield pretty much identical identifications of SSW, as well as their classification into type-1 and type-2 SSW;

[Figure]

2. That the EP flux divergence makes the most important contribution to w* at high latitudes as determined from a generalized version of "downward control"; and

2a. That the resulting warming of the polar cap occurs due to "advection" [actually adiabatic warming] by the induced TEM circulation.

Neither of these findings is surprising and the second is definitely not new. One can find a similar description in Andrews et al (1987, and references therein). Some discussion of the role of gravity wave drag is presented, but found to be small. There is a weak attempt to relate the findings to the work of Albers and Birner (JAS, 2014) but it is unconvincing. All in all, there appears to be no change in this version of the paper compared to the version submitted for preliminary review. Therefore, I am afraid I still cannot recommend the paper for publication in ACP.

Specific comments (page and line number):

(1,32) "waves are broken": "waves break" would be preferred. That aside, wave breaking need not be the only dissipation mechanism in the case of planetary Rossby waves, whose group velocity is slow enough that they can be affected by thermal dissipation. In fact, Matsuno (1971) never discussed wave breaking; that concept came much later, with McIntyre and Palmer's paper in Nature (1983).

(3,12) Section 3: The material on the downward control (DC) definitions of v*, w* logically should follow what is now section 3.2, since the DC equations are derived from the TEM set presented in 3.2. In addition, Eqs. (1)-(4) should be moved after Eq. (9) in what is now Sec. 3.2, since they are definitions of terms in that equation. Section 3 should be reorganized, such that the present Section 3.2 becomes Section 3.1 and the material on DC becomes a short Section 3.2.

(5,7) "For Type-1 (composite mean)...": It should be mentioned here that the composites are shown on the two lower, RHS panels of Fig. 1.

(5,39) "Figure 3 shows...": Here and in the figure caption you need to note that what

is shown are composites for Type 1 and Type 2 (and their difference).

(6,16) "A large proportion": From Fig. 3, GWD is -2 to -5 m/s, but EPFD is -10 to -20 m/s, so GWD is at most 20-25% of EPFD, and that only near 1 hPa. I would not consider this a "large" fraction. Perhaps you should just state the numbers and let the reader decide.

(6,40) "The GWD forcing anomaly has different structures . . .": I don't quite see this. If one compares 3b and 3d one sees similar behavior: Not much contribution before the key date (lag=0) and a positive contribution after the key date. What is remarkable here is how different this looks compared to the original composites (without subtracting climatology). One interpretation of this is that GWD does not differ much from its climatological value before the key date, but after the key date GWD is suppressed, such that the difference from climatology is positive. This behavior is consistent with the idea that the reversal of the wind inhibits GW propagation and thus reduces the GWD.

(7,13) "Although the magnitude . . .": This is stated without proof and is unconvincing. It is not at all clear from what is shown here that the small forcing due to GWD is important for the generation of SSW. One could equally argue that GWD is responding to the underlying zonal-mean zonal wind, which has been modified due to other causes.

(7,42) "temperature advection": This is contributed mainly by $w^*S$ ($S$ = kappa T/H), so it is actually adiabatic warming or cooling due to vertical motion. As is well known, this is the principal mechanism whereby a sudden warming warms the polar stratosphere.

(8,13) "adiabatic heating": I believe you mean "diabatic" (third column of Figure 6). If so, note that this is really a response to the temperature change brought about by dynamics (adiabatic effects–what you call "advection"). It is not a driver of the SSW.

(9,16) "To summarize": This is the main finding of the paper, but it is neither new nor surprising. And, once again, "anomalous cooling" is a response (IR relaxation) to the

temperature changes that accompany the sudden warming.

(9,24) "results . . . not specific to just one data set": This is useful to know but not particularly surprising insofar as all of the reanalyses ultimately rely on the same observational data.

(10.,3) "EPD is the most significant contribution. . .": Again, this is hardly news.
* * *

---

## Referee Comment (RC2) · Anonymous Referee #3 · 14 Dec 2016

This study uses a modern reanalysis (MERRA) to compute and composite forcing terms in the transformed Eulerian mean (TEM) zonal wind and thermodynamic equations about sudden stratospheric warmings (SSWs). The authors separate the SSWs into Type-1 and Type-2 events based on relative sizes of wave-1 and wave-2 polar geopotential height anomalies. The composites demonstrate that the planetary-scale wave activity flux is the dominant forcing term in both the TEM zonal wind and thermodynamic equations. Uppermost stratospheric gravity wave drag and middle stratospheric diabatic heating are meanwhile shown to be small, though non-negligible. I believe the authors present a clean analysis that stays on point with the paper's theme.

[Figure]

There are however a few points that I'd like the authors to edit or address to boost the quality of the manuscript.

Principally, I'm not sure what the added value of separating the events into two types is. While comparison of the two types seems to be a large portion of the analysis, there is not much discussion on the implications of these results. Events are often separated in studies of SSWs, but the reasons need to be made clear. I don't believe the authors have amply done this in the introduction or summary. I think a more thorough discussion of why the authors did what they did and how it fits into the literature will greatly aid the manuscript.

There are also a few analysis steps by which the authors could address this problem. Firstly, the authors could show an 'all SSW' composite for each part of the analysis. In this way, the manuscript would analyze the residual mean circulation from MERRA in all SSWs and concurrently show the results for one way that SSWs are separated.

Secondly, the authors could (and I believe should) show significance of the anomalies for each event type. While the significant difference between Type-1 and Type-2 is important, so too is their own significant difference from zero. Especially given the small sample size, this will better inform the reader as to which composite structures agree with each other.

Given the scope and work required for these changes, I recommend that the manuscript be returned for major revisions.

Minor comments:

Page 2, line 23: I think you should state that it has not been done with the generalized downward control principle.

Page 2, line 41: how is the climatology calculated? This will be important information if others wish to reproduce or adapt your results.

On reproducibility, thank you for including a table of SSW dates. This step is often

overlooked for SSW studies.

Page 5, line 42: what is the reason you average the forcing over a different latitude band than the residual forcing terms?

Page 7, line 19: though the amplitude may be small, the residual term has a broad region of significant difference. Do the authors have any insight as to why that may be?

Page 9, line 21: model level data from ERA-Interim is used, but Table 1 indicates pressure-level data is used (i.e., shouldn't ERA-Interim have 60 levels?).

On the figures: since so many panels are included in each figure, the panels will be quite small when published. This will make seeing the small regions of significance hard to see. I'm not sure the best way to do this, but the authors may consider altering their figures to better show hatched regions. This is especially true over regions with dark blue contour filling.

---

## Author Comment (AC1) · 22 Feb 2017

**Response to Short Comments**

Thank you for providing valuable comments that improve the original manuscript. We tried our best to improve the manuscript based on your suggestions.

**General comments:**

*Dear authors,*

*I would like to ask you to extend your analysis range in Fig. 4a to approx. 30˚N to capture the possible GWD anomalies "above Japan" (see the EA/NP hotspot in Šácha et al. (2015) and its possible implications in Šácha et al. (2016)). However, this hotspot is dominant in the vertical range between 30hPa and 10hPa, which is below your analysis range. Please consider as well to supplement your existing analysis (10hPa-1hPa, where the Scandinavia hotspot is dominant) with two additional plots of GWD anomalies between e.g. 30hPa and 10hPa.*

*Best wishes, Petr Šácha.*

→ As suggested, the GWD anomalies averaged between 30 and 10 hPa for Type-1 (left) and Type-2 (right) SSW events at Lag = −15 are calculated and shown in Fig. A1. We found that there is no significant GWD anomaly in the east Asian-northwestern Pacific (EA/NP; 37.5° N–62.5° N, 112.5° E–168.8° E) region (denoted by box in Fig. A1), where the hotspot of the gravity wave potential energy exists in Šácha et al. (2015). This discrepancy is interesting and worth to be investigated as a future research topic. As suggested, Fig. A1 is included as a supplement figure in the revised manuscript. [Page 7, line 29–34]

[Figure]

Figure A1. Polar-stereo projection maps of the GWD anomaly averaged in height between 30 and 10 hPa for Type-1 (left) and Type-2 (right) SSW events in the 15 days before the central date. The red box denotes the EA/NP (37.5° N –62.5° N, 112.5° E –168.8° E) region.

**References**:

Šácha, P., Kuchař, A., Jacobi, C., and Pišoft, P.: Enhanced internal gravity wave activity and breaking over the northeastern Pacific–eastern Asian region, Atmos. Chem. Phys., 15, 13097-13112, doi:10.5194/acp-15-13097-2015, 2015.

---

## Author Comment (AC2) · 22 Feb 2017

**Response to Reviewer #1's Comments**

Thank you for providing valuable comments that improve the original manuscript. We tried our best to improve the manuscript based on your suggestions.

**Major comments:**

*This paper examines SSW in several reanalysis datasets and reaches two main conclusions:*

*1. That the various datasets yield pretty much identical identifications of SSW, as well as their classification into type-1 and type-2 SSW;*

*2. That the EP flux divergence makes the most important contribution to w\* at high latitudes as determined from a generalized version of "downward control"; and*

*2a. That the resulting warming of the polar cap occurs due to "advection" [actually adiabatic warming] by the induced TEM circulation.*

*Neither of these findings is surprising and the second is definitely not new. One can find a similar description in Andrews et al (1987, and references therein). Some discussion of the role of gravity wave drag is presented, but found to be small. There is a weak attempt to relate the findings to the work of Albers and Birner (JAS, 2014) but it is unconvincing. All in all, there appears to be no change in this version of the paper compared to the version submitted for preliminary review. Therefore, I am afraid I still cannot recommend the paper for publication in ACP.*

→ As we know, this is the first study that examines the contribution of each wave forcing to the temperature change during the evolution of SSWs, based on the generalized downward control principle.

During the revision process, the relative magnitude of GWD to total wave forcing (EPD+GWD) is calculated and a new result is included as Fig. 4 in the revised manuscript. It is found that GWD/(EPD+GWD) averaged in 60° N–70° N is up to 90% in the upper stratosphere before warming, especially for the Type-2 cases. After Lag = 0, there are several heights and times of which GWD/(EPD+GWD) is more than 50% in the whole stratosphere for both cases. This implies that contribution of GWD to SSW is rather large locally. [Page 6, line 28–33]

**Minor comments:**

1) *(1,32) "waves are broken": "waves break" would be preferred. That aside, wave breaking need not be the only dissipation mechanism in the case of planetary Rossby waves, whose group velocity is slow enough that they can be affected by thermal dissipation. In fact, Matsuno (1971) never discussed wave breaking; that concept came much later, with McIntyre and Palmer's paper in Nature (1983).*

→ Thank you for pointing out this. We modify the sentence to "critical layer interaction", which was presented in the Matsuno (1971). [Page 1, line 33–34]

2) *(3,12) Section 3: The material on the downward control (DC) definitions of v\*, w\* logically should follow what is now section 3.2, since the DC equations are derived from the TEM set presented in 3.2. In addition, Eqs. (1)-(4) should be moved after Eq. (9) in what is now Sec. 3.2, since they are definitions of terms in that equation. Section 3 should be reorganized, such that the present Section 3.2 becomes Section 3.1 and the material on DC becomes a short Section 3.2.*

→ We agree with you, and it is reorganized, as suggested. [Page 3, line 16–page 4, line 27]

3) *(5,7) "For Type-1 (composite mean)…": It should be mentioned here that the composites are shown on the two lower, RHS panels of Fig. 1.*

→ We already mentioned this in the original manuscript. [Page 5, line 3] [Page 5, line 15 of the revised manuscript]

4) *(5,39) "Figure 3 shows…": Here and in the figure caption you need to note that what is shown are composites for Type 1 and Type 2 (and their difference).*

→ It is modified in the revised manuscript, as suggested. [Page 6, line 9–13 and page 20, line 1]

5) *(6,16) "A large proportion": From Fig. 3, GWD is -2 to -5 m/s, but EPFD is -10 to -20 m/s, so GWD is at most 20-25% of EPFD, and that only near 1 hPa. I would not consider this a "large" fraction. Perhaps you should just state the numbers and let the reader decide.*

→ To quantify the fraction of GWD, percentages of GWD to the total wave forcing (EPD+GWD) for Type-1 and Type-2 SSWs are calculated during the revision process and shown in a new figure (Fig. 4) of the revised manuscript. From Lag = –15 to –10, GWD contributes more than 20% of the total wave forcing above 10 hPa in both types of SSWs, and high percentage (> 70%) is also observed, especially for Type 2. This is consistent with Fig. 2 of Albers and Birner (2014) which shows that the percentage of GWD to the total wave forcing above 10 hPa in the 60˚ N–70˚ N is about 20–80% at Lag –30 to –5. [Page 6, line 28–33]

6) *(6,40) "The GWD forcing anomaly has different structures…": I don't quite see this. If one compares 3b and 3d one sees similar behavior: Not much contribution before the key date (lag=0) and a positive contribution after the key date. What is remarkable here is how different this looks compared to the original composites (without subtracting climatology). One interpretation of this is that GWD does not differ much from its climatological value before the key date, but after the key date GWD is suppressed, such that the difference from climatology is positive. This behavior is consistent with the idea that the reversal of the wind inhibits GW propagation and thus reduces the GWD.*

→ As we mentioned in the original manuscript, the difference between the GWD anomaly for two types of SSW is particularly large at Lag = –15 in the upper stratosphere, where statistical significant positive (Type-1) and negative (Type-2) anomalies exist. As shown in Fig. 3e, the difference between the two types of anomaly fields also significantly large. Therefore, it can be said that GWD anomaly for two types of SSW has different

structures. When the lag time extends to ±25 days from the original ±15 days, we found (Fig. A1) significant negative GWD anomalies in the upper stratosphere between Lag = −25 days and −15 days exclusively for Type-2. Because it is hard to see the contours and arrows clearly, if we extend the time series, we decided to keep the current lag time (Lag = ±15 days).

[Figure]

Figure A1. Time-height cross section of GWD anomalies (shading) averaged over 60° N–70° N and the residual mean velocity anomalies (red arrow) averaged over 70° N–80° N, induced by GWD for (a) Type-1 and (b) Type-2 SSW events. The hatch patterns denote statistical significance at 90% confidence level.

7) (7,13) "Although the magnitude…": This is stated without proof and is unconvincing. It is not at all clear from what is shown here that the small forcing due to GWD is important for the generation of SSW. One could equally argue that GWD is responding to the underlying zonal-mean zonal wind, which has been modified due to other causes.

→ We agree with the reviewer's criticism, and delete this statement in the revised manuscript. Although we are currently working on the indirect effect of GWs on SSW, it is not proper to state in the current manuscript without concrete results.

8) (7,42) "temperature advection": This is contributed mainly by w*S (S = kappa T/H), so it is actually adiabatic warming or cooling due to vertical motion. As is well known, this is the principal mechanism whereby a sudden warming warms the polar stratosphere.

→ Yes, it is correct. The adiabatic warming/cooling rate term is dominant among the terms included in the "temperature advection" term in the original manuscript. To avoid any confusion, "temperature advection" is changed to "adiabatic heating rate" in the revised manuscript.

9) (8,13) "adiabatic heating": I believe you mean "diabatic" (third column of Figure 6). If so, note that this is really a response to the temperature change brought about by dynamics (adiabatic effects–what you call "advection"). It is not a driver of the SSW.

→ Thank you for pointing out this typo error. Yes, it is diabatic heating. As we know, the anomalous diabatic heating is a forcing to determine the temperature change, and it is a forcing to derive zonal-mean zonal wind tendency and residual circulation [see Eq. (3.5.7)

and (3.5.8) of Andrew et al. (1987)]. Figures 9 and 10 show that diabatic heating has a similar structure (cooling in poleward of 60° N) before and after the warming, while its contribution to the temperature change differs as the adiabatic heating changes dramatically before and after the warming. It is not simply a response to the temperature change accompany the sudden warming. If our understanding is not correct, please let us know. We will make a further revision regarding this part latter.

10) (9,16) "To summarize": This is the main finding of the paper, but it is neither new nor surprising. And, once again, "anomalous cooling" is a response (IR relaxation) to the temperature changes that accompany the sudden warming.

→ Many part of the results, such as dominancy of EPD forcing in TEM equation during SSWs, are similar to the previous studies, which cannot be different as long as our analyses are correct. As mentioned earlier, however, this is the first study on the contribution of each wave forcing to the temperature change during the evolution of SSW, based on the generalized downward control principle. In addition, as answered to the previous comment (#9), we do not think that the diabatic heating considered here is the response of SSW but the forcing to contribute to temperature change during the evolution of SSW.

11) (9,24) "results … not specific to just one data set": This is useful to know but not particularly surprising insofar as all of the reanalyses ultimately rely on the same observational data.

→ Yes, it is correct that recent reanalysis data sets can produce similar structure of the large-scale wind and temperature fields, in general, based on similar observational data. However, it is worth to check the robustness of the current results based on MERRA with different data sets, mainly because small differences in wind and temperature can induce significant differences in wave forcing terms and residual circulations induced by the wave forcing, which are calculated from high-order derivatives. This point is included in the revised manuscript. [Page 9, line 41–43]

12) (10,3) "EPD is the most significant contribution…": Again, this is hardly news.

→ Again, the current result is consistent with previous ones, although we used GWD information as well, which is not commonly used in previous studies. The magnitude of GWD provided from MERRA is not larger than EPD. However, as shown in the new figure (Fig. 4), the ratio of GWD to total wave forcing (EPD+GWD) is up to 90% at some heights and times before the warming, especially for the Type-2 SSWs, and this implies that GWD is a non-negligible forcing for driving for SSWs. The statement is modified based on the new figure.

**Reference:**

Albers, J. R., and Birner, T.: Vortex preconditioning due to planetary and gravity waves prior to sudden stratospheric warmings. J. Atmos. Sci., 71, 4028–4054, doi: 10.1175/JAS-D-14-0026.1, 2014.

Andrews, D. G., Holton, J. R. and Leovy, C. B.: Middle Atmosphere Dynamics. Academic Press, 489 pp, 1987.

Matsuno, T.: A dynamical model of stratospheric warmings. J. Atmos. Sci., 28, 1479–1494, doi: 10.1175/1520-0469(1971)028<1479:ADMOTS>2.0.CO;2, 1971.

---

## Author Comment (AC3) · 22 Feb 2017

**Response to Reviewer #3's Comments**

Thank you for providing valuable comments that improve the original manuscript. We tried our best to improve the manuscript based on your suggestions.

**Major comments:**

1) *This study uses a modern reanalysis (MERRA) to compute and composite forcing terms in the transformed Eulerian mean (TEM) zonal wind and thermodynamic equations about sudden stratospheric warmings (SSWs). The authors separate the SSWs into Type-1 and Type-2 events based on relative sizes of wave-1 and wave-2 polar geopotential height anomalies. The composites demonstrate that the planetary-scale wave activity flux is the dominant forcing term in both the TEM zonal wind and thermodynamic equations. Uppermost stratospheric gravity wave drag and middle stratospheric diabatic heating are meanwhile shown to be small, though non-negligible. I believe the authors present a clean analysis that stays on point with the paper's theme.*

2) *There are however a few points that I'd like the authors to edit or address to boost the quality of the manuscript. Principally, I'm not sure what the added value of separating the events into two types is. While comparison of the two types seems to be a large portion of the analysis, there is not much discussion on the implications of these results. Events are often separated in studies of SSWs, but the reasons need to be made clear. I don't believe the authors have amply done this in the introduction or summary. I think a more thorough discussion of why the authors did what they did and how it fits into the literature will greatly aid the manuscript. There are also a few analysis steps by which the authors could address this problem. Firstly, the authors could show an 'all SSW' composite for each part of the analysis. In this way, the manuscript would analyze the residual mean circulation from MERRA in all SSWs and concurrently show the results for one way that SSWs are separated. Secondly, the authors could (and I believe should) show significance of the anomalies for each event type. While the significant difference between Type-1 and Type-2 is important, so too is their own significant difference from zero. Especially given the small sample size, this will better inform the reader as to which composite structures agree with each other. Given the scope and work required for these changes, I recommend that the manuscript be returned for major revisions.*

→ We classify SSWs into two types, as they have distinct characteristics in wave dynamics (Charlton and Polvani 2007; Nishii et al., 2011), impact on the weather (O'Callaghan, 2014; Liu et al., 2014; Seviour et al., 2015), and gravity waves drag (Albers and Birner, 2014; Šácha et al., 2016). Following the reviewer's suggestion, reasons to separate the SSW events into two types are included in the revised manuscript [page 2, line 2–5] with the references.

Following the reviewer suggestion, some analyses in the original manuscript are re-performed for all SSW cases during the revision process. The result (Fig. A1) of the all SSW cases is about the average of the result by the two types, as expected. Therefore, the result is not included in the revised manuscript.

Following the reviewer's suggestion, statistical significance of the anomalies for each SSW types is checked in all figures of the original manuscript, and significant anomalies

are highlighted by hatch patterns in Fig. 3b, 3d, 5a, 6b, 7a, 7b, 8a, 8b, 9a, 9b, 10a, 10b, and 11 of the revised manuscript.

[Figure]

[Figure]

Figure A1. Time-height cross sections of the composites of (a) each forcing term averaged over 60° N–70° N (shading) and (b) adiabatic heating rate by the residual mean circulation averaged over 70° N–80° N (shading). The arrows denote the residual mean velocities averaged over 70° N–80° N, induced by each forcing. (c) Time-height cross sections of the all SSWs composites of the zonal-mean temperature tendency (first column), adiabatic heating rate by the circulation (second column), diabatic heating rate (third column), eddy heat flux (fourth column), and the residual term of TEM thermodynamic energy equation (fifth column) averaged over 70° N–80° N. The first to third rows of each panel are composites for all, Type-1, and Type-2 SSW events, respectively. All values are anomaly fields from the climatology. The hatch patterns denote statistically significance at 90% confidence level.

**Minor comments:**

1) *Page 2, line 23: I think you should state that it has not been done with the generalized downward control principle.*

→ It is modified, as suggested. [Page 2, line 25]

2) *Page 2, line 41: how is the climatology calculated? This will be important information if others wish to reproduce or adapt your results.*

→ The climatology is defined as the 34-year average of each day. This is included in the revised manuscript. [Page 2, line 42]

3) *On reproducibility, thank you for including a table of SSW dates. This step is often overlooked for SSW studies.*

→ Thank you for pointing out this. We hope that this information helps other scientists to study SSW.

4) *Page 5, line 42: what is the reason you average the forcing over a different latitude band than the residual forcing terms?*

→ Here, we said not "residual forcing" but "residual circulation" ($\bar{v}^*$, $\bar{w}^*$) induced by the forcing. If there is negative forcing at midlatitude, say 60° N, it induces a poleward and downward motion on latitudes higher, say 70° N, than the forcing latitude. Therefore, we used the different latitude band for wave forcing and residual circulation.

5) *Page 7, line 19: though the amplitude may be small, the residual term has a broad region of significant difference. Do the authors have any insight as to why that may be?*

→ The test statistic for two-sample t-test (Wilks 1995) is determined by Eq. (A1):

$$t = \frac{\bar{X}_1 - \bar{X}_2}{\sqrt{\frac{s_1^2}{N_1} + \frac{s_2^2}{N_2}}} \ , \tag{A1}$$

where $\bar{X}$, $s^2$, and N are the sample mean, variance, and size, respectively. For the case of the residual term between Type 1 and Type 2, we found that the test statistic is large due to the relatively small standard deviation, regardless of relative small mean values. This is included in the revised manuscript. [Page 7, line 36–37]

6) *Page 9, line 21: model level data from ERA-Interim is used, but Table 1 indicates pressure-level data is used (i.e., shouldn't ERA-Interim have 60 levels?).*

→ Thank you for pointing out this. When SSWs are selected, we used four reanalysis data sets including ERA-Interim pressure-level data. The ERA-Interim model-level data are used only for Fig. 11. The information of the model level data of ERA-Interim is added to Table 1 of the revised manuscript, along with a statement related to usage of ERA-Interim pressure-level and model-level data sets mentioned above. [Page 14, Table 1]

*7) On the figures: since so many panels are included in each figure, the panels will be quite small when published. This will make seeing the small regions of significance hard to see. I'm not sure the best way to do this, but the authors may consider altering their figures to better show hatched regions. This is especially true over regions with dark blue contour filling.*

→ We agree with you. Although we have tried hard to make figures more visible, we could not find a satisfactory way, in order to keep all necessary panels in a figure. To make the hatched regions in the figures more clearly visible, the colors of the contour label in Fig. 3 are changed.

**References**:

Albers, J. R., and Birner, T.: Vortex preconditioning due to planetary and gravity waves prior to sudden stratospheric warmings. J. Atmos. Sci., 71, 4028–4054, doi: 10.1175/JAS-D-14-0026.1, 2014.

Charlton, A. J., and Polvani, L. M.: A new look at stratospheric sudden warmings. Part I: Climatology and modeling benchmarks. J. Clim., 20, 449–469, doi: 10.1175/JCLI3996.1, 2007.

Liu, C., Tian, B., Li, K.-F., Manney, G. L., Livesey, N. J., Yung, Y. L., and Waliser, D. E.: Northern Hemisphere mid-winter vortex-displacement and vortex-split stratospheric sudden warmings: Influence of the Madden-Julian Oscillation and Quasi-Biennial Oscillation, J. Geophys. Res. Atmos., 119, 12, 599–12,620, doi:10.1002/2014JD021876, 2014.

Nishii, K., Nakamura, H., and Orsolini, Y.: Geographical Dependence Observed in Blocking High Influence on the Stratospheric Variability through Enhancement and Suppression of Upward Planetary-Wave Propagation. J. Climate, 24, 6408–6423, doi: 10.1175/JCLI-D-10-05021.1, 2011.

O'Callaghan, A., Joshi, M., Stevens, D., and Mitchell, D.: The effects of different sudden stratospheric warming type on the ocean, Geophys. Res. Lett., 41, 7739–7745, doi:10.1002/2014GL062179, 2014.

Šácha, P., Lilienthal, F., Jacobi, C., and Pišoft, P.: Influence of the spatial distribution of gravity wave activity on the middle atmospheric dynamics, Atmos. Chem. Phys., 16, 15755-15775, doi:10.5194/acp-16-15755-2016, 2016.

Seviour, W. J. M., Gray, L. J., and Mitchell, D. M.: Stratospheric polar vortex splits and displacements in the high-top CMIP5 climate models, J. Geophys. Res. Atmos., 121, 1400–1413, doi:10.1002/2015JD024178, 2016.

Wilks, D. S.: Statistical methods in the atmospheric sciences, second edition. Academic Press, 464 pp, 1995.